# Prediction of Durability of Thermal Insulating Epoxy Coatings with Regard to Climatic Ageing

**DOI:** 10.3390/polym14091650

**Published:** 2022-04-19

**Authors:** Zhanar O. Zhumadilova, Vladimir P. Selyaev, Ruslan E. Nurlybayev, Yelzhan S. Orynbekov, Indira B. Sangulova, Erzhan I. Kuldeyev

**Affiliations:** 1Institute of Architecture and Civil Engineering, Satbayev University, 22a Satpaev St., Almaty 050013, Kazakhstan; i.sangulova@stud.satbayev.university (I.B.S.); e.kuldeyev@satbayev.university (E.I.K.); 2Department of Building Constructions, Ogarev Mordovia State University, 68 Bolshevitsk St., 430005 Saransk, Russia; ntorm80@mail.ru; 3LLP SAVENERGY Industrial Innovation Company, Industrial Zone, Almaty 050013, Kazakhstan; nurlybayev.savenergy@gmail.com (R.E.N.); orynbekov.savenergy@gmail.com (Y.S.O.)

**Keywords:** epoxy, hardener, microspheres, composite, climate, polymer, coatings, color, climate factor

## Abstract

It is generally accepted that the color and performance characteristics of liquid thermal insulation coatings are affected by the combined effect of various climatic factors such as solar radiation, temperature fluctuations, moisture, precipitation and others. This work presents the results of a scientific study of the full-scale exposure of coatings with regard to climatic ageing. Methods have been carried out, such as: spectrophotometry and direct scanning; determining adhesion, determining the adhesion strength of facing and protective coatings; and thermal conductivity and thermal resistance. As the results of the research work have shown, only in situ climatic tests, accompanied by the obligatory recording of the aggressive factors affecting the coating, make it possible to assess changes in the properties of epoxy coatings in full-scale conditions and, consequently, their climatic resistance by the methods of spectrophotometry and direct scanning. The ageing of polymer composites is known to be accompanied by a change not only in elasticity but also in color. Of the epoxy coatings tested, Etal-45M showed the greatest color variation during the in situ climate test. The most decorative resistant coatings are obtained using epoxy resin ED-20 + modified epoxy resin Etal-1440N.

## 1. Introduction

Energy saving is the most important task of our time. Many countries in Europe, the USA and Russia have adopted energy conservation programs and laws. Energy consumption in the European Union is to be reduced by 50% by 2050 compared to 1990 (European Commission 2008).

It has been established that about 30% of the resources extracted and created are lost in heating buildings, transporting heat carriers.

New design solutions, technologies and materials are proposed to solve the energy saving problem. Liquid thermal insulation coatings (LTIC) are attracting increasing research attention [1,2,3,4]. They make it possible to create monolithic, seamless coatings for complex surface configurations.

Liquid thermal insulation coatings (LTIC) are an energy-saving material for application on pipelines, metal and concrete surfaces, bricks, plaster, etc., with unique thermal conductivity properties superior to traditionally used thermal insulation materials. The composition of the LTIC includes (depending on modification): polymeric dispersion (based on acrylic copolymers and polyurethane binders), glass, ceramic or aluminosilicate microspheres, and ultra- and nanosized powders (microsilica, metakaolin, diatomite, white carbon black, aerosil, etc.). The product has been designed to be used with a range of special additives and additions (thickeners, dispersants, defoamers, pigments, biocides, etc.).

The use of microsilica, glass, ceramic and aluminosilicate microspheres and finely dispersed mineral fillers in these coatings makes it possible to:Improve thermal insulation properties;Reduce the weight of the coatings;Increase the heat and light reflectivity of the coatings.

The low values of the thermal conductivity coefficient of such compositions are usually ensured by the introduction of a large number of hollow ceramic and glass microspheres into the composition of the LTIC, up to 80% of the total volume, according to literature sources. The role of the binder in the additional reduction of the thermal conductivity of the formulations is hardly taken into account.

Epoxy binder coatings are the most widely used in construction practices because of their versatile properties [4,5]. There is known experience in applying epoxy coatings: to protect concrete and metal from aggressive media [3,6,7,8,9,10]; as decoration [11,12]; for heat insulation [3,13]; and to further reinforce reinforced concrete structures [5].

Epoxy coatings are multifunctional and to increase their resistance to heat transfer it is necessary to increase the content volume of filler like MS (MS—microspheres, hollow glass or alumosilicates microspheres).

The aim of the proposed work is to analyze changes in the conditions and to develop methods for predicting the durability of epoxy coatings, which are decorative, thermal insulation and protection and are operated under the influence of climatic factors—moisture, temperature, and solar radiation.

The scientific novelty of the work lies in the proposed method of predicting the durability of epoxy coatings based on a non-destructive method of color-change control.

Assessing the influence of climatic factors on the properties of a polymer composite is a complex task, as only some of the following can be evaluated in full-scale operating conditions: color, gloss, hardness, and surface condition [14,15,16,17].

The evaluation of the physical-chemical and mechanical properties of epoxy resins can be estimated from the following works of the authors:At work [18], the results showed that the functionalization and chemical compatibility of APTES-treated MWCNTs with an epoxy composition provides increases of the physical and mechanical properties of epoxy composites: bending stress increases by 194% and bending modulus increases by 137%; the tensile strength increases by 108% and the tensile elastic modulus increases by 52%; impact strength increases by 300%.At work [19] in this study, the mechanical properties of the composites were tested by changing particle size, volume fraction and temperature. The mechanical properties of the composites were improved by optimizing the fabrication conditions by controlling the viscosity of the composites.

Depending on the physical and chemical properties, epoxy resin (ED) grades were established and selected.

Resins, hardeners, plasticizers, solvents whose production is mastered in the Russian Federation and their composition is regulated by the corresponding Governmental Standards (for example, epoxy resin ED-20 (GOST 10587-84) [20].

Recently, Etal-1441 complex hardeners and Etal-247 epoxy binders have appeared on the market. The aim is to assess the quality of the coatings produced on the basis of these components in terms of their physical and mechanical properties.

Using experimental data on the mechanical and chemical properties of the coating material under laboratory conditions, as well as the fundamental laws of physical chemistry, it is possible to predict the quantitative values of the material characteristics at any given time [21].

## 2. Materials and Methods

### 2.1. Materials

Samples for experimental studies were made on the basis of epoxy compositions consisting of:*Epoxy resin ED-20 (epoxy-dian resin)*—dynamic viscosity at (25 ± 0.1) °C—12–25 Pa·s, gelatinization time at least 4 h, mass fraction of epoxy groups at least 20–22.5%; Etal-247 is a low-viscosity modified epoxy resin. Viscosity 20–22 times lower than ED-20, gelatinization time at least 1.5 h, mass fraction of epoxy groups at least 21.4–22.8%;*Hardener Etal-1440N* is a low-viscosity two-pack epoxy compound, viable for up to 3 h, curing period 24 h, at 20–25 °C, workable up to 110 °C after additional heat treatment; Etal-1460 amine type is a homogenous low-viscous liquid of light to dark brown color, designed for the curing of epoxy resins and compounds based on them at 0–+40 °C in any humidity and under water, curing time of compound at 0–10 °C is 24 h, at 10–20 °C is 16 h; Etal-45M is an amine type hardener used to harden epoxy resins and compounds based on them at temperatures from −7 °C (recommended from +5) to +45 °C, % humidity is not important; polyethylene polyamine (PEPA) is a cold-curing hardener for epoxy resins and compositions. It is a mixture of ethylene amines. PEPA is fully soluble in water and in alcohol. Plasticizer-dibutyl phthalate (DBPT), coal tar—(CT); diluents-butanol; fillers-microsilica (McS) and microspheres (MS); antioxidants (AO)-salicylic and phenolic acids.

When selecting a composition for protective and decorative coatings with high resistance to climatic influences, preliminary (test) tests were carried out to determine the effect of each component on coating properties.

### 2.2. Methods

Nature testing was carried out using the automatic control station of Ogarev Mordovia State University (Saransk). The samples were placed on test benches in the environmental and metrology laboratory. Measurements of metrological parameters (temperature, relative air humidity, total solar radiation, ultraviolet (UV) radiation, atmospheric pressure, wind speed and direction, precipitation and pollutant content) were taken automatically every 20 min around the clock [3].

The analysis of variation in decorative characteristics was carried out with a CD-6834 spectro-guide sphere gloss spectrophotometer, as well as with a direct scanning method using the personal computer “Statistical analysis of color components of paintwork coatings”. According to the results of the studies for each composition, after a set period of time, the color coordinates of the test sample were determined to be *L_T_*^∗^, *a_T_*^∗^, *b_T_*^∗^ and the color coordinates of the comparison sample (control sample before climatic exposure) to be *L_R_*^∗^, *a_R_*^∗^, *b_R_*^∗^.

When tested under laboratory conditions, the specimens were manufactured by applying the epoxy composite to a substrate of dense polyethylene, which was detached after curing. Stencils were used to make samples of a given shape and size.

The adhesion strength was determined by the tear-off method using a PSO MG4 adhesion meter. The essence of the method is to determine the ultimate resistance of the cured coating to detachment from the substrate. The bond strength (adhesion) is determined by the peeling force of the cured coating sample from the substrate applied to the sample through a metal disc with an anchor bonded to the surface of the sample. As a sample of the test coating, a disk with a diameter of 30 mm was used, the tear force was increased and the force was increased at a rate of not more than 1 MPa/s, perpendicular to the plane of the surface to be painted, so that destruction occurred within 90 s from the beginning of the application of the tear force.

The thermal conductivity coefficient was determined by the steady-state heat flux method using the ITS-1. The principle of operation is based on the creation of a stationary heat flux through the flat specimen in question. The thermal conductivity of the sample λ is automatically calculated from the value of this heat flux, the temperature of the opposite faces of the sample and the thickness of the sample:–Thermal conductivity measurement range—0.02–1.5 W/(m·K);–Thermal resistance measurement range—0.01–1.5 m^2^·K/W;–The limits of permissible relative error in measuring thermal conductivity and thermal resistance are ±5%;–Thickness of the measured sample—10–25 mm.

The mechanical tests for the determination of strength and deformation modulus were determined using an AGS-X, TRAPEZIUMX tensile testing machine [3].

For each composition, the color coordinates of the test sample *L_T_*^∗^, *a_T_*^∗^, *b_T_*^∗^ and the color coordinates of the comparison sample (control sample before climatic exposure) *L_R_*^∗^, *a_R_*^∗^, *b_R_*^∗^ were determined after a set time period.

The total color difference ∆*E_ab_*^∗^ between the two colors was calculated from the values obtained Equation (1):∆*E_ab_*^∗^ = [(∆*L*^∗^)^2^ + (∆*a*^∗^)^2^ + (∆*b*^∗^)^2^]^1/2^(1)
where ∆*L*^∗^ = *L_T_*^∗^ − *L_R_*^∗^; ∆*a*^∗^ = *a_T_*^∗^ − *a_R_*^∗^; ∆*b*^∗^ = *b_T_*^∗^ − *b_R_*^∗^, *L_T_*^∗^, *a_T_*^∗^, *b_T_*^∗^—color coordinates of the test sample, *L_R_*^∗^, *a_R_*^∗^, *b_R_*^∗^—color coordinates of the comparison sample; difference in color tone ∆*H_ab_*^∗^ = *k*_*H*_ {[(∆*E_ab_*^∗^)^2^ − (∆*L_ab_*^∗^)^2^ − (∆*C_ab_*^∗^)^2^]^1/2^}.

Direct scanning method undertaken using a personal computer “Statistical analysis of the color components of paintwork coatings” [3].

The samples were scanned using a 9000F Mark II full-color flatbed scanner at 2400 *dpi* in CMYK (cyan, magenta, yellow, key or black) format mode.

For the control samples, the total color saturation including luminance was determined according to Equations (2) and (3):(2)$$E_{CMYKH}=\sqrt{S_{C}^{2}+S_{M}^{2}+S_{\gamma }^{2}+S_{K}^{2}}$$
where *S_C_*, *S_M_*, *S_Y_*, *S_K_* and *S_H_*—color differences in saturation for cyan, magenta, yellow, black and brightness respectively, compared to absolute white Equation (3):(3)$$S_{p}=\frac{\sum_{i=0}^{255}\left(255-X_{pi}\right)\cdot f\left(X_{pi}\right)}{255\cdot 100}$$
where $X_{pi}$—color component level, varying from 0 to 255; $f\left(X_{pi}\right)$—density of the distribution.

The methodology for preparing epoxy coatings is standard for paint materials:1)Dosing of epoxy resin, hardener, and microspheres and thinner, using a laboratory scale;2)Mix the components in a laboratory mixer. Mixing is done at 60–70 rpm for 10–15 min.3)The resulting composition is passed through a strainer and fed to the filler. To obtain a high-quality coating, it is necessary to ensure a high level of physical and mechanical properties (hiding power, density and layer thickness). It is therefore recommended that for primed surfaces the material consumption should not exceed 400 g/m^2^ for the first coat. For rough surfaces the consumption rate for the first coat is increased by 20–30%.

## 3. Results

### 3.1. Test Results

It is known that the decorative properties of coatings are commonly evaluated according to four indicators: gloss change, color change, dirt retention and chalking. Protective properties are evaluated according to five indicators: cracking, weathering (erosion), peeling, degradation of coating material and substrate [3].

Epoxy coatings are comparatively expensive and their use is reasonable for the purpose of protecting products from external influences. Therefore, cracking and degradation have been taken as the main test indicators, which are evaluated by tensile strength σ and modulus of elasticity *E* [4].

Table 1 shows the values for the dynamic modulus of elasticity in GOST R 57862 (government standard). The modulus of elasticity in static compression will be one order of magnitude lower [22].

The essence of the method is to induce oscillations by applying a periodic force to the sample, directed along its axis. The corresponding fundamental resonant frequencies, sample dimensions and masses are used to calculate the dynamic modulus of elasticity, dynamic shear modulus and Poisson’s ratio.

Compositions 1 and 2 will be the most sensitive to the effects of climatic factors, as the analysis of experimental data [5] shows that in the first composition there are high internal stresses, while in the second composition dibutyl phthalate desorbs (sweats out) during operation and reduces the protective and decorative properties.

The most resistant compositions 5 and 6 will be those in which internal stresses are reduced by the filler and plasticizer.

For comparison purposes, the Etal compositions offered by the current market were tested. In some cases, they are hardeners (Etal 1440N); in others, they are binders (Etal 247). These formulations are similar to formulations 5 and 6 in terms of their components.

### 3.2. The Experimental Research Results

The study of changes in the properties of epoxy coatings during exposure on the test benches of the Ogarev Mordovia State University (Saransk) climatic station was carried out with the compositions given in Table 2.

It has been observed during research that the greatest impact of UV radiation on polymer composites is reflected in their decorative properties. Changes in a polymer’s surface appearance indicate the speed of the degradation processes caused by aggressive factors.

The color coordinates of the test piece, *L_T_*^∗^, *a_T_*^∗^, *b_T_*^∗^, and the color coordinates of the comparison piece (control piece before climatic exposure), *L_R_*^∗^, *a_R_*^∗^, *b_R_*^∗^, were calculated from the following color differences.

The graphical relationship between the change in total color difference and the color tone difference of the polymer coatings obtained with the spectrophotometer is shown in Figure 1, Figure 2, Figure 3 and Figure 4. These graphs have been grouped according to the type of resin part, allowing the effect of the hardener type on the change in the decorative performance of the polymers to be clearly traced.

The analysis of the graphs in Figure 1, Figure 2, Figure 3, Figure 4 and Figure 5 shows that the strength, mass, ultimate strains and tensile modulus of elasticity change under the influence of climatic factors. Changes in strength and modulus of elasticity are due to changes in the structure of the material.

In the course of long-term climatic exposure, a process of damage accumulation develops in the composite structure (see graph 5 in Figure 5). The change in the complete color difference of epoxy coatings during exposure to nature is also due to the accumulation of micro-fractures, damage to the material structure. Graph 5 (in Figure 5) is therefore similar to the graphs in Figure 1, Figure 2, Figure 3 and Figure 4. Consequently, the durability of coatings can be assessed and predicted, both on the basis of the strength change data and the color difference data. For the mathematical description of the degradation process of epoxy coatings we will use the partial damage superposition principle, using the Bailey criterion (Equation (4)):(4)$$\overset{n}{\underset{i=1}{\sum }}\frac{\Delta t_{i}}{\tau \left(x\right)}$$
where $\Delta t_{i}$—duration of energy exposure (voltage, radiation, etc.); $\tau \left(x\right)$—durability of the composite (time to failure) under the action of a destructive factor x–stress; corrosive environment; humidity; radiation.

In [5,23] it is assumed that the failure of polymer composites is due to the accumulation of elementary damage and the kinetic process of failure can be described by an Equation (5) of the form:(5)$$\tau =\tau_{0}exp\left\{\left(\frac{u_{0-\gamma \delta \sigma -\beta Q}}{kT}\right)\right\}$$
where τ—time to destruction (fracture); $\tau_{0}$—molecular vibration period; σ—tension (stress); δ—fracture criterion; γ, β—constants;$\;u_{0}$—activation energy; k—Boltzmann’s constant; T—temperature, $k^{0}$; Q—radiation energy; σ and Q—function of time *t*.

Solving Equations (4) and (5) together, a mathematical model can be derived to describe changes in color (*E*) and strength (*σ*), under the influence of climatic factors as a function (Equation (6)):(6)$$\Delta P\left(x\right)=\alpha_{x}X_{i}^{\beta_{x}}\times exp\left(\frac{-\beta_{x}X_{i}}{\gamma_{x}}\right)\leq P_{n}$$
where the coefficients *α*, *β,* and *γ* are determined from experimental data; *P_n_* is the normative structure damage index.

From the analysis of the data obtained, it was found that for all three types of resins used, the greatest color difference ∆*E_ab_*^∗^ (Figure 1 and Figure 2) and color tone difference ∆*H_ab_*^∗^ (Figure 3 and Figure 4) was observed with the Etal-45M hardener.

At the same time, for the polymers based on epoxy resin + Etal-247, after reaching the maximum values at the level of 210–240 days, a rather sharp decrease in color differences is observed (Figure 2). This time interval, corresponding to the month of July, is characterized by the maximum values of the actinometrical parameters, as well as the greatest surface heating of the samples under study. The numerical values of the coefficients of Equation (2) for total color difference ∆*E_ab_*^∗^ and color tone difference ∆*H_ab_*∗ are given in Table 3 and Table 4, respectively.

Determination of the coefficient values, ranging from 0.72 to 0.98 for the full color difference, indicate a sufficiently high reliability of the models obtained. The values 113 *R*^2^ for ∆*H_ab_*^∗^ depending on composition, time and actinometrical parameter (*Q*, *U*_A_ and *U*_B_) vary in the range 0.79 ÷ 0.96 for all compositions except (Etal-247), which has the least change in decorative properties.

In parallel with determining the change in the decorative parameters of the epoxy polymers, change was determined in strength, tensile modulus, relative elongation, mass, degree of damage to the structure, and adhesive strength of the thermal conductivity coefficient during climatic tests. Typical graphs are shown in Figure 5 [24,25].

The following data were obtained from work carried out to determine the durability of epoxy coatings with regard to climatic ageing.

Graphical dependencies describing the variation in decorative values were obtained using the personal computer “Statistical analysis of the color components of paintwork coatings” [3]. When using the direct-scanning method, similar trends in color differences during exposure to nature are observed, as is the case with damage and color, so it was concluded that it was reasonable to use the same relationship to process the data obtained from Equation (6). The numerical values of mathematical model Equation (6) for the total color difference ∆*E_CMYKH_* are shown in Table 5.

## 4. Discussion


Coatings based on formulations cured with Etal-45M are characterized by the greatest color variation during 10 months of exposure to the natural climate.The proposed mathematical model makes it possible to reliably describe the process of change in the decorative properties of epoxy coatings depending on the duration of field exposure, total solar radiation and ultraviolet A and B radiations.The most decorative resistant coatings are obtained using epoxy resin ED-20 + modified epoxy resin Etal-1440N. The recommended coating composition for the epoxy coating groups is shown in Table 6.



The use of the personal computer “Statistical analysis of color components of paintwork coatings” does not require expensive equipment (scans can be obtained by means of a common flatbed scanner), which makes its application the most expedient for the evaluation of the colorimetric characteristics of epoxy polymers and the paintwork coatings based on them. Given the critical values not perceived by the human eye (∆*E_ab_*^∗^ ≤ 2.75), a similar parameter ∆*E_CMYKH_* must not exceed 0.18 when using the direct-scanning method.In the process of research, it was found that under the influence of ultraviolet radiation, a process of micro-damage accumulation develops in the polymer coating, which adequately affects the color and durability of the coating.A mathematical model, Equation (6), describing the degradation process of polymer coatings based on experimental data of color difference change was developed.


## 5. Conclusions

As the results of the joint research work between Satbayev University (Kazakhstan) and Ogarev Mordovia State University (Saransk) in the framework of an international project collaboration have shown, in spite of certain difficulties and the duration of the research, only climatic tests, accompanied by the obligatory recording of the aggressive factors affecting the coating, allow a full assessment of the changes in the properties occurring in full-scale conditions and, consequently, their climatic durability. The need to quantify climatic factors demonstrates the particular relevance of using spectrophotometry and the personal computer “Statistical analysis of color constituents of paintwork coatings” for this purpose.

The analysis of changes in the polymer sample characteristics after exposure to climatic factors shows that the compositions based on ED-20 epoxy cured with amine hardeners Etal-1440N show the best elasticity and decorative resistance characteristics after 10 months of testing. The high stability of the values considered leads to the conclusion that the use of resin + Etal as a base leads to the most weather-resistant epoxy coatings.

According to the results of the test tests (Section 3.1), it was determined that the selected epoxy composites most resistant to external influences are compositions 5 and 6, as shown in Table 7, in which internal stresses are reduced by a filler and a plasticizer. Epoxy composites are in some cases hardeners (Etal-1440N) and in other cases they are binders (Etal-247).

## Figures and Tables

**Figure 1 polymers-14-01650-f001:**
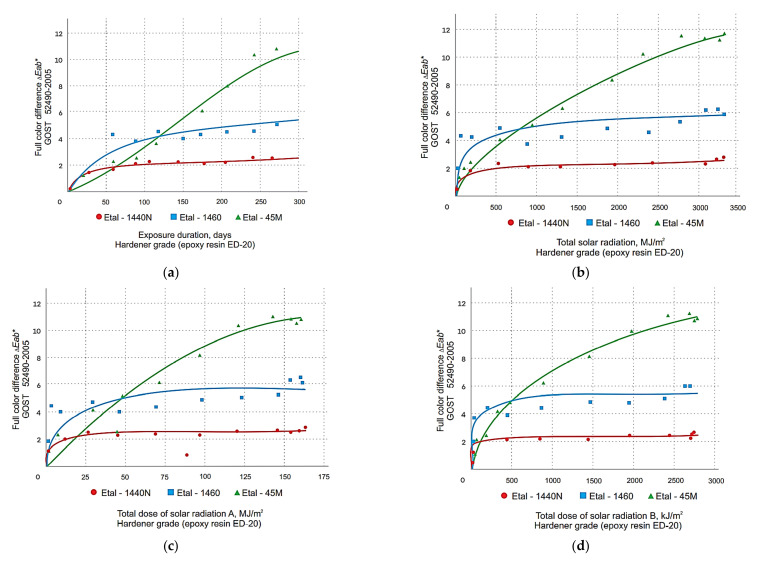
Change in the total color difference of polymers based on epoxy resin ED-20 depending on the duration of exposure (**a**), total solar radiation (**b**), total ultraviolet radiation of the range A (**c**) and B (**d**) when exposed to climatic factors.

**Figure 2 polymers-14-01650-f002:**
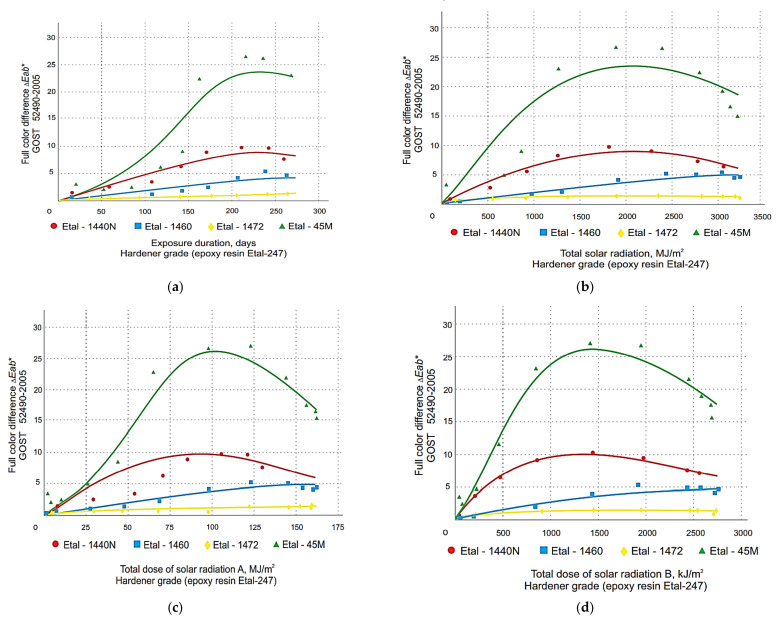
Change in the total color difference of polymers based on the modified Etal-247 epoxy resin depending on the exposure time (**a**), total solar radiation (**b**), total ultraviolet radiation of the range A (**c**) and B (**d**) when exposed under the influence of climatic factors.

**Figure 3 polymers-14-01650-f003:**
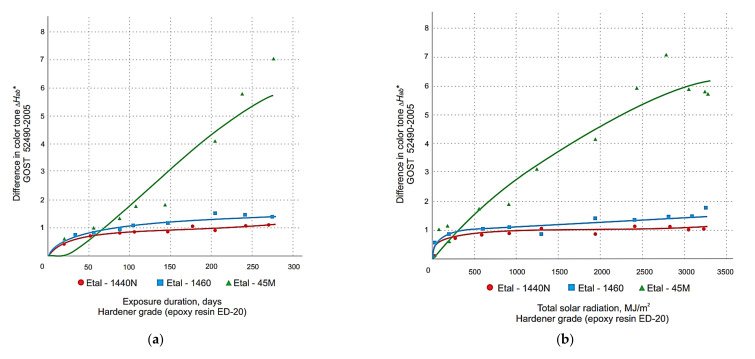
Change in the difference in the color tone of polymers based on epoxy resin ED-20 depending on the duration of exposure (**a**), total solar radiation (**b**), total ultraviolet radiation of the range A (**c**) and B (**d**) when exposed under the influence of climatic factors.

**Figure 4 polymers-14-01650-f004:**
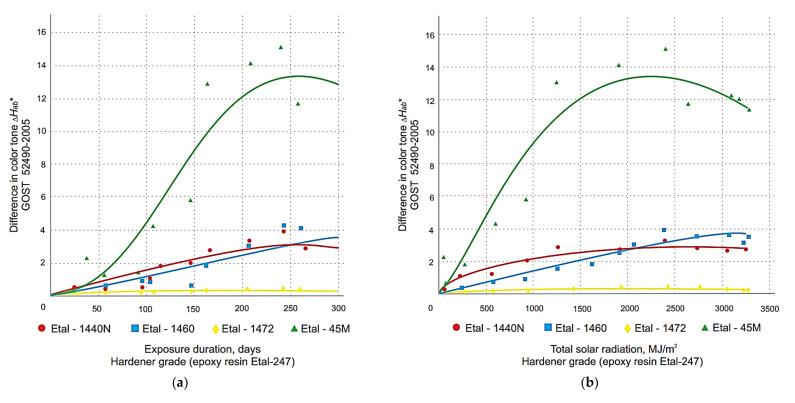
Change in the difference in the color tone of polymers based on the modified epoxy resin Etal-247 depending on the duration of exposure (**a**), total solar radiation (**b**), total ultraviolet radiation of the range A (**c**) and B (**d**) when exposed under the influence of climatic factors.

**Figure 5 polymers-14-01650-f005:**
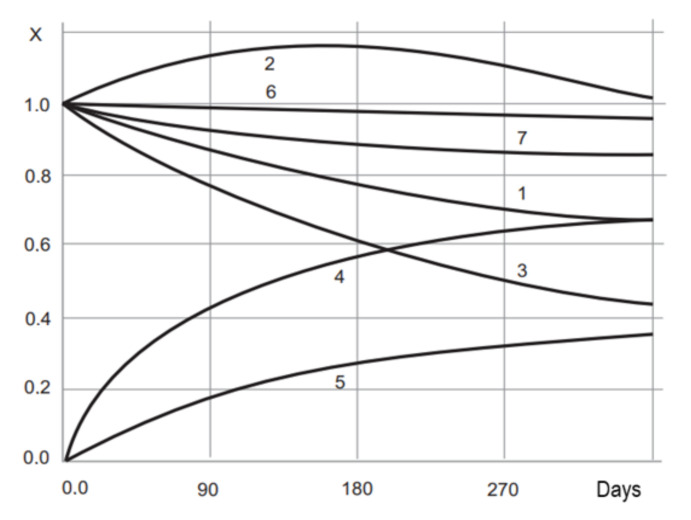
Change (x); strength (**1**); tensile modulus (**2**); elongation (**3**); mass (**4**); structure damage (**5**) of epoxy composition (composition 6 of Table 1); adhesion strength (**6**); thermal conductivity (**7**); during climatic testing.

**Table 1 polymers-14-01650-t001:** Test compositions (in mass proportions) of epoxy compositions.

No.	Epoxy Resin ED-20	Polyethylene Polyamine (PEPA)	Dibutyl Phthalate (DBPT)	$\frac{\mathrm{McS}}{\mathrm{MS}}$ Microsilica/Microspheres	Antioxidants (AO)	Butanol	*σ*, MPa	*E*, MPa
1	100	10	-	-	-	-	21	$2\times 10^{4}$
2	100	10	10	-	-	-	20	$1.5\times 10^{4}$
3	100	10	10	$\frac{50}{25}$	-	-	29	$2.5\times 10^{4}$
4	100	10	-	-	-	10	22	$1.8\times 10^{4}$
5	100	10	10	$\frac{50}{25}$	2	10	28	$3\times 10^{4}$
6	100	10	10	$\frac{50}{25}$	2	10	25	$3\times 10^{4}$

**Table 2 polymers-14-01650-t002:** Compositions of epoxy coatings exhibited on coal tar (CT) stands.

No.	Epoxy Resin Grade	Hardener Grade	Antioxidants (AO)	Diluent
1	Epoxy resin ED–20	Polyethylene polyamine (PEPA)	-	-
2	Epoxy resin ED–20	Modified epoxy resin Etal–1440N	-	-
3	Modified epoxy resin Etal–247	Polyethylene polyamine (PEPA)	-	-
4	Epoxy resin ED-20	Polyethylene polyamine (PEPA)	Phenolic acid	Butanol

**Table 3 polymers-14-01650-t003:** Values of the coefficients of Equation (2) describing the influence of exposure time (*T*), total solar radiation intensity (*Q*) and *A* (*U*_A_) and *B* (*U*_B_) on the total color difference of epoxy polymers exposed under full-scale conditions (spectrophotometer).

Compound Number	Hardener Type	Considered Factor	The Values of the Coefficients of Equation (2)			
			α_x_	β_x_	γ_x_	R^2^
**Epoxy Resin ED-20**						
1	Modified epoxy resin Etal–1440N	T, days	8.22 × 10^−1^	1.99 × 10^−1^	37,954	0.947
		Q, MJ/m^2^	1.16	9.80 × 10^−2^	31,989	0.932
		U_A_, MJ/m^2^	1.56	9.60 × 10^−2^	7914	0.932
		U_B_, kJ/m^2^	1.82	3.90 × 10^−2^	56,088	0.904
2	Modified epoxy resin Etal–1460	T, days	3.20 × 10^−1^	6.03 × 10^−1^	312.8	0.860
		Q, MJ/m^2^	1.52	1.66 × 10^−1^	110,990	0.865
		U_A_, MJ/m^2^	1.20	4.02 × 10^−1^	127	0.724
		U_B_, kJ/m^2^	2.97	8.10 × 10^−2^	35,699	0.878
3	Modified epoxy resin Etal–45M	T, days	1.42 × 10^−3^	1.80	401	0.974
		Q, MJ/m^2^	4.61 × 10^−2^	7.19 × 10^−1^	7072	0.978
		U_A_, MJ/m^2^	3.70 × 10^−2^	1.37	172	0.940
		U_B_, kJ/m^2^	6.86 × 10^−2^	7.10 × 10^−1^	3547	0.946
**Epoxy Resin Etal–247**						
4	Modified epoxy resin Etal–1440N	T, days	6.00 × 10^−4^	2.12	242	0.844
		Q, MJ/m^2^	5.00 × 10^−4^	1.52	1758	0.955
		U_A_, MJ/m^2^	2.50 × 10^−3^	2.37	89.3	0.973
		U_B_, kJ/m^2^	3.60 × 10^−3^	1.29	1240	0.972

**Table 4 polymers-14-01650-t004:** The values of the coefficients of Equation (2), which describes the effect of the exposure time (*T*), the intensity of total solar radiation (*Q*) and ultraviolet radiation of the A (*U*_A_) and B (*U*_B_) ranges on the difference in the color tone of the epoxy polymers exposed to natural conditions (spectrophotometer).

Compound Number	Hardener Type	Considered Factor	The Values of the Coefficients of Equation (2)			
			α_x_	β_x_	γ_x_	R^2^
**Epoxy Resin ED-20**						
1	Modified epoxy resin Etal–1440N	T, days	1.92 × 10^−1^	3.14 × 10^−1^	38,292	0.802
		Q, MJ/m^2^	3.03 × 10^−1^	1.67 × 10^−1^	65,902	0.807
		U_A_, MJ/m^2^	4.98 × 10^−1^	1.66 × 10^−1^	11,801	0.809
		U_B_, kJ/m^2^	5.88 × 10^−1^	8.3 × 10^−2^	40,498	0.824
2	Modified epoxy resin Etal–1460	T, days	1.32 × 10^−1^	4.4 × 10^−1^	1240	0.851
		Q, MJ/m^2^	2.88 × 10^−1^	2.07 × 10^−1^	35,998	0.853
		U_A_, MJ/m^2^	5.37 × 10^−1^	2.03 × 10^−1^	7948	0.853
		U_B_, kJ/m^2^	6.70 × 10^−1^	9.8 × 10^−2^	38,798	0.787
3	Modified epoxy resin Etal–45M	T, days	4.00 × 10^−5^	2.44	363	0.924
		Q, MJ/m^2^	1.46 × 10^−3^	1.12	4858	0.937
		U_A_, MJ/m^2^	1.79 × 10^−2^	1.35	206.1	0.937
		U_B_, kJ/m^2^	6.86 × 10^−2^	5.81 × 10^−1^	16,810	0.941
**Epoxy Resin Etal-247**						
4	Modified epoxy resin Etal–1440N	T, days	8.00 × 10^−4^	1.77	292	0.913
		Q, MJ/m^2^	1.84 × 10^−2^	7.48 × 10^−1^	2608	0.929
		U_A_, MJ/m^2^	4.47 × 10^−2^	1.15	111.5	0.919
		U_B_, kJ/m^2^	4.00 × 10^−2^	6.87 × 10^−1^	1636	0.904

**Table 5 polymers-14-01650-t005:** Values of the coefficients of Equation (2), which describes the effect of the exposure time (*T*), the intensity of total solar radiation (*Q*) and ultraviolet radiation of the A (*U*_A_) and B (*U*_B_) ranges on the total color difference of the epoxy polymers exposed to natural climatic conditions (direct scan method).

Compound Number	Hardener Type	Considered Factor	The Values of the Coefficients of Equation (2)			
			α_x_	β_x_	γ_x_	R^2^
**Epoxy Resin ED-20**						
1	Modified epoxy resin Etal–1440N	T, days	6.49 × 10^−3^	5.58 × 10^−1^	513	0.375
		Q, MJ/m^2^	1.74 × 10^−2^	2.45 × 10^−1^	8282	0.773
		U_A_, MJ/m^2^	3.63 × 10^−2^	2.41 × 10^−1^	481	0.773
		U_B_, kJ/m^2^	4.29 × 10^−2^	1.27 × 10^−1^	13,719	0.810
2	Modified epoxy resin Etal–1460	T, days	2.15 × 10^−2^	3.94 × 10^−1^	1591	0.768
		Q, MJ/m^2^	4.08 × 10^−2^	1.97 × 10^−1^	150,189	0.744
		U_A_, MJ/m^2^	6.93 × 10^−2^	2.08 × 10^−1^	9596	0.743
		U_B_, kJ/m^2^	7.43 × 10^−2^	1.26 × 10^−1^	150,458	0.740
3	Modified epoxy resin Etal–45M	T, days	9.93 × 10^−3^	5.94 × 10^−1^	298	0.744
		Q, MJ/m^2^	3.65 × 10^−2^	2.04 × 10^−1^	2471	0.700
		U_A_, MJ/m^2^	5.56 × 10^−2^	2.79 × 10^−1^	116	0.712
		U_B_, kJ/m^2^	8.55 × 10^−2^	9.20 × 10^−2^	1575	0.716
**Epoxy resin Etal-247**						
4	Modified epoxy resin Etal–1440N	T, days	4.00 × 10^−4^	1.34	352	0.946
		Q, MJ/m^2^	3.00 × 10^−4^	0.1	2130	0.962
		U_A_, MJ/m^2^	4.70 × 10^−3^	1.10	109	0.954
		U_B_, kJ/m^2^	2.20 × 10^−4^	1.16	1360	0.871

**Table 6 polymers-14-01650-t006:** Recommended compositions.

No.	Compositions
1	Epoxy resin ED-20 + modified epoxy resin Etal-1440N ratio (100:56)
2	Microspheres-20%
3	Thinner-from application technology

**Table 7 polymers-14-01650-t007:** Test compositions (by weight, %) of epoxy compositions.

Compound Number	Epoxy Resin ED-20	Polyethylene Polyamine (PEPA)	Dibutyl Phthalate (DBPT)	$\frac{\mathrm{McS}}{\mathrm{MS}}$ Microsilica/Microspheres	Antioxidants (AO)	Butanol	*σ*, MPa	*E*, MPa
5	100	10	10	$\frac{50}{25}$	2	10	28	$3\times 10^{4}$
6	100	10	10	$\frac{50}{25}$	2	10	25	$3\times 10^{4}$

## Data Availability

The data presented in this study are available on request from the corresponding author.

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
