# Peer review of "Prediction of Durability of Thermal Insulating Epoxy Coatings with Regard to Climatic Ageing"

_polymers, 2022, doi:10.3390/polym14091650_

Round 1

Reviewer 1 Report

The manuscript reported the prediction of the durability of epoxy coatings in long-term servicing under the influence of climatic factors. The research topic is undoubtedly important both for the academic and industrial aspects. However, the current manuscript needed to be majorly revised so as to meet the requirements for publication.

  1. The chemical expressions are woefully inadequate in the manuscript. Effects of the chemical compositions of the epoxy coating on the final properties are critically important. Without such information, one cannot justify the adaptability and accuracy of the experimental and predicting results. For example:

1) The elastic modulus (E) values of the epoxy thermosetting coatings are unreasonable and questionable. For the common epoxy/amine curing system without any additives, the E values over dozens of GPa (2*104 MPa, or 20 GPa) in Table 1 is nearly impossible. Please find the literature, such as Journal of Applied Polymer Science, 2007, 106(3): 2047-2055.

2) Line 76, the detailed expression for the starting materials should be given, such as the commercial resources of the compounds, the epoxy value or epoxide equivalent weight (EEW) of the epoxy resin, the type of the epoxy resin, the amine values of the curing agents, and so on. Polyethylene polyamine (PEPA) is a class of amine-type curing agents, which one does the author use? What is the type of epoxy resin, bisphenol A type or other type?

3) Line 85, the experimental procedure for preparing the epoxy/amine thermosetting coatings should be given.

  1. Line 85, in the Methods part, please provide the predicting software and corresponding theory except the CIE Lab color parameters. The authors only present the CIE Lab parameters. These are not the predicting methods. Corresponding expression in 3.2 part should be moved to 2.2 part.
  2. Line 143 and line 299, Table 1 and Table 7 should be cited in the main text part in the manuscript, respectively.
  3. Line 143, in Table 1, the units for the E were suggested to be “GPa” instead of “MPa”.
  4. Some grammatical and spelling mistakes should be revised, such as: 1) line 79 and line 47, the misuse of the brackets; 2) line 121, the misuse of the letters, and so on.

Author Response

Response to Reviewer 1 Comments

Dear Reviewer!

Thank you for your consideration of our manuscript “Prediction of Durability of Thermal Insulating Epoxy Coatings with Regard to Climatic Ageing”. 

Point 1: The chemical expressions are woefully inadequate in the manuscript. Effects of the chemical compositions of the epoxy coating on the final properties are critically important. Without such information, one cannot justify the adaptability and accuracy of the experimental and predicting results. For example:

Response 1: Yes, we agree. The following texts have been added to the work.

The evaluation of the physical-chemical and mechanical properties of epoxy resins can be estimated from the following works of the authors:

  • At work [11] the results showed that the functionalization and chemical compatibility of APTES-treated MWCNTs with epoxy composition provides increased of physical and mechanical properties of epoxy composites: bending stress increases by 194% and bending modulus increases by 137%, the tensile strength increases by 108% and the tensile elastic modulus increases by 52%, impact strength increases by 300%.
  • At work [12] in this study, the mechanical properties of composite were tested by changing particle size, volume fraction and temperature. The mechanical properties of the composites were improved by optimizing the fabrication conditions by controlling the viscosity of the composites.

Depending on the physical and chemical properties, epoxy resin (ED) grades were established and selected.

Resins, hardeners, plasticizers, solvents whose production is mastered in the Russian Federation and their composition is regulated by the corresponding Governmental Standards (for example, epoxy resin ED-20 (GOST 10587-84) [18].

Recently, Etal-1441 complex hardeners and Etal-247 epoxy binders have appeared on the market. The aim is to assess the quality of the coatings produced on the basis of these components in terms of their physical and mechanical properties.

Point 2: The elastic modulus (E) values of the epoxy thermosetting coatings are unreasonable and questionable. For the common epoxy/amine curing system without any additives, the E values over dozens of GPa (2*104 MPa, or 20 GPa) in Table 1 is nearly impossible. Please find the literature, such as Journal of Applied Polymer Science, 2007, 106(3): 2047-2055.

Response 2: Table 1 shows the values of the dynamic modulus of elasticity in GOST R 57862. The modulus of elasticity in static compression will be 1 order of magnitude lower. The essence of the method is to induce oscillations by applying a periodic force along the samples axis. The corresponding fundamental resonance frequencies, dimensions and mass of the sample are used to calculate the dynamic modulus of elasticity, dynamic shear modulus and Poisson's coefficient.

Point 3: Line 76, the detailed expression for the starting materials should be given, such as the commercial resources of the compounds, the epoxy value or epoxide equivalent weight (EEW) of the epoxy resin, the type of the epoxy resin, the amine values of the curing agents, and so on. Polyethylene polyamine (PEPA) is a class of amine-type curing agents, which one does the author use? What is the type of epoxy resin, bisphenol A type or other type?

Response 3:

  1. The compositions of Table 1 were used in the pre-test tests.
  2. After preliminary tests, the epoxy coatings were determined, the composition of which is given in Еables 2-6.

Point 4: Line 85, the experimental procedure for preparing the epoxy/amine thermosetting coatings should be given.

Response 4: The methodology for preparing epoxy coatings is standard for paint materials:

1) Dosing of epoxy resin, hardener, and microspheres and thinner, using a laboratory scale;

2) Mix the components in a laboratory mixer. Mixing is done at 60-70 rpm for 10-15 min.

3) The resulting composition is passed through a strainer and fed to the filler. To obtain a high quality coating, it is necessary to ensure a high level of physical and mechanical properties (hiding power, density, and layer thickness). It is therefore recommended that for primed surfaces the material consumption should not exceed 400 g/m2 for the first coat. For rough surfaces the consumption rate for the first coat is increased by 20-30 %.

Point 5: Line 85, in the Methods part, please provide the predicting software and corresponding theory except the CIE Lab color parameters. The authors only present the CIE Lab parameters. These are not the predicting methods. Corresponding expression in 3.2 part should be moved to 2.2 part.

Response 5: The transfer of part of the text according to the method of item 3.2 to item 2.2 has been carried out.

Point 6: Line 143 and line 299, Table 1 and Table 7 should be cited in the main text part in the manuscript, respectively.

Response 6: Table 1 and Table 7 in the text are percented.

Point 7: Line 143, in Table 1, the units for the E were suggested to be “GPa” instead of “MPa”.

Response 7: Table 1 originally proposed the unit of measure E in MPa.

Point 8: Some grammatical and spelling mistakes should be revised, such as: 1) line 79 and line 47, the misuse of the brackets; 2) line 121, the misuse of the letters, and so on.

Response 8: Agreed, all lines corrected.

Best Regards,

Authors group

Reviewer 2 Report

The manuscript under the title: “Prediction of Durability of Thermal Insulating Epoxy Coatings with Regard to Climatic Ageing” is in line with Polymers journal. This topic is relevant and will be of interest to the readers of the journal. It based on original research. This research has scientific novelty and practical significance. The article has a typical organization for research articles.
Before the publication it requires significant improvements, especially:

1. The "Introduction" section needs to be substantially reworked. It is necessary to expand the review of the literature on the use of epoxy composites as heat-insulating materials - to identify the main trends in their modification in order to improve the heat-shielding properties.

2. It has been proven that the effect of fillers on the properties of polymer composites is determined by many factors: ……. I think the related references should be cited corresponding to each aspect, e.g. (but not limited to these), which will undoubtedly improve the "Introduction" section:

  • Polymers 2020, 12 (8), 1816, https://doi.org/10.3390/polym12081816
  • Polymers 2020, 12(1), 79, https://doi.org/10.3390/polym12010079
  • International Polymer Science and Technology. 2013, V. 40, N. 7, P. 49-51. https://doi.org/10.1177/0307174X1304000711
  • Polym. Compos. 2019, 40, 3877–3885. https://doi.org/10.1002/pc.25246

3. What is the scientific novelty of your work? This should be detailed in the Introduction section.

4. Section 2.1. It is necessary to add the physicochemical characteristics of all components - give a table with the main physicochemical and technological properties of all components. In addition, the manufacturer and country of production of all chemicals must be indicated.

5. Section 2.2. all standards according to which all determinations of the properties of composites were carried out should be indicated.

6. Section 2.2. it is necessary to indicate the dimensions and number of samples for each type of test.

7. GOST standards should be replaced by their corresponding international ISO standards.

8. What is the difference between compositions 5 and 6 (Table 1 and 7)?

9. In all tables, all symbols and abbreviations must be deciphered.

10. Section 4. Item 3 and data in Table 6. do not match the data in tables 1 and 7. Why?

11. What is the ratio of fillers MeS and MS in the compositions. And what was the rationale for choosing this ratio?

12. Formulas 5-7 and their description I recommend moving to section 2.2.

13. Discussion: I recommend expanding this section substantially. Please compare achieved results with up-to-date literature, also with composites with other admixtures. Discuss the achieved results.

14. What is the purpose of table 7? Where is the link to it in the text?

Author Response

Response to Reviewer 2 Comments

Dear Reviewer!

Thank you for your consideration and comments to our manuscript “Prediction of Durability of Thermal Insulating Epoxy Coatings with Regard to Climatic Ageing”. 

Point 1: The "Introduction" section needs to be substantially reworked. It is necessary to expand the review of the literature on the use of epoxy composites as heat-insulating materials - to identify the main trends in their modification in order to improve the heat-shielding properties.

Response 1: Epoxy coatings are multifunctional and to increase resistance to heat transfer it is necessary to increase the volume content of filler like MS (MS – microspheres, hollow glass or alumosilicates microspheres).

Point 2: It has been proven that the effect of fillers on the properties of polymer composites is determined by many factors: ……. I think the related references should be cited corresponding to each aspect, e.g. (but not limited to these), which will undoubtedly improve the "Introduction" section:

  • Polymers 2020, 12 (8), 1816, https://doi.org/10.3390/polym12081816
  • Polymers 2020, 12(1), 79, https://doi.org/10.3390/polym12010079
  • International Polymer Science and Technology. 2013, V. 40, N. 7, P.

49-51. https://doi.org/10.1177/0307174X1304000711

  • Compos.2019, 40, 3877–3885. https://doi.org/10.1002/pc.25246

Response 2: Yes, we agree.

The evaluation of the physical-chemical and mechanical properties of epoxy resins can be estimated from the following works of the authors:

  • At work [11] the results showed that the functionalization and chemical compatibility of APTES-treated MWCNTs with epoxy composition provides increased of physical and mechanical properties of epoxy composites: bending stress increases by 194% and bending modulus increases by 137%, the tensile strength increases by 108% and the tensile elastic modulus increases by 52%, impact strength increases by 300%.
  • At work [12] in this study, the mechanical properties of composite were tested by changing particle size, volume fraction and temperature. The mechanical properties of the composites were improved by optimizing the fabrication conditions by controlling the viscosity of the composites.

Depending on the physical and chemical properties, epoxy resin (ED) grades were established and selected.

Resins, hardeners, plasticizers, solvents whose production is mastered in the Russian Federation and their composition is regulated by the corresponding Governmental Standards (for example, epoxy resin ED-20 (GOST 10587-84) [18].

Recently, Etal-1441 complex hardeners and Etal-247 epoxy binders have appeared on the market. The aim is to assess the quality of the coatings produced on the basis of these components in terms of their physical and mechanical properties.

Point 3: What is the scientific novelty of your work? This should be detailed in the Introduction section.

Response 3: The scientific novelty of the work lies in the proposed method of predicting the durability of epoxy coatings based on a non-destructive method of color change control.

Point 4: Section 2.1. It is necessary to add the physicochemical characteristics of all components - give a table with the main physicochemical and technological properties of all components. In addition, the manufacturer and country of production of all chemicals must be indicated.

Response 4: The components that make up the coatings are too varied in their chemical composition (for example, epoxy resin ED-20 and MS) and properties, so they cannot be combined in one table.

Point 5: Section 2.2. all standards according to which all determinations of the properties of composites were carried out should be indicated.

Response 5: In addition to the text, GOSTs are given:

GOST (Government Standard) 10587-84. Epoxy-dian uncured resins. Technical conditions, Moscow: Publishing house of standards, 1989. 20p.

GOST (Government Standard) R 57862 – 2017. Composites. Determination of dynamic modulus of elasticity, shear modulus and Poisson's ratio by acoustic resonance, Moscow: Standardinform, 2019. 13p.

GOST 32299-2013 (ISO 4624: 2002). Paint and varnish materials. Determination of adhesion by tear-off method. Moscow: Standartinform, 2014. 15p.

GOST (Government Standard) 7076-99. Materials and articles of construction. Method for determination of thermal conductivity and thermal resistance under steady-state thermal conditions. Moscow: Standartinform, 2000. 27p.

Point 6: Section 2.2. it is necessary to indicate the dimensions and number of samples for each type of test.

Response 6: The number of samples in Tableы 1, 2 and 7 are presented. In Tables 3,4 and 5, the data of the number of samples added. It is difficult to indicate the dimensions of the samples, since according to GOST, the test methods are different.

Point 7: GOST standards should be replaced by their corresponding international ISO standards.

Response 7: Replacing GOST standards with ISO is not possible, because not all GOSTs have an analogue of ISO.

According to ISO corresponds to:

GOST R 52489- 2005 (ISO 7724-1:1984).

GOST R 52490-2005 (ISO 7724-3:1984).

GOST 32299-2013 (ISO 4624: 2002).

Point 8: What is the difference between compositions 5 and 6 (Table 1 and 7)?

Response 8: Tables 1 and 7 show preliminary evaluation tests for cracking and degradation of epoxy coatings. Tables 5 and 6 show the result of the final composition of the epoxy coatings.

Point 9: In all tables, all symbols and abbreviations must be deciphered.

Response 9: All tables have been deciphered.

Point 10: Section 4. Item 3 and data in Table 6. do not match the data in tables 1 and 7. Why?

Response 10: Tables 1 and 7 show preliminary evaluation tests for cracking and degradation of epoxy coatings. Table 6 presents the result of the final composition of epoxy coatings.

Point 11: What is the ratio of fillers MeS and MS in the compositions. And what was the rationale for choosing this ratio?

Response 11: The ratio of McS and MS components is 1:0.5. (for 1 part of the resin 0.5 parts of the microsphere). Table 1 and 7 corrected.

Point 12: Formulas 5-7 and their description I recommend moving to section 2.2.

Response 12: Formulas 5-7 are tied to figures and general text.

Point 13: Discussion: I recommend expanding this section substantially. Please compare achieved results with up-to-date literature, also with composites with other admixtures. Discuss the achieved results.

Response 13: Added:

  1. In the process of research, it was found that under the influence of ultraviolet radiation, a process of micro-damage accumulation develops in the polymer coating, which adequately affects the color and durability of the coating.
  2. A mathematical model (Eq. 7) describing the degradation process of polymer coatings based on experimental data of color difference change was developed.

In the Results and Discussion section, we presented the results of our own work and discuss the results obtained from our own research, so we do not cite other authors here.

Point 14: What is the purpose of table 7? Where is the link to it in the text?

Response 14: The link in the text is made to Table 7.

In the Table 7, preliminary test compositions (by weight, %) of epoxy compositions are the most resistant to the action of climatic factors.

After preliminary tests, the types of epoxy coatings were determined, the composition of which is presented in Tables 2-6.

Best Regards,

Authors group

Reviewer 3 Report

In the Methods part, explain methods in more detail. For example for, tensile testing add speed at which samples were tested, as well as, samples geometry and number of samples.

Line 144 place ab in subscript in ∆Eab∗

Line 145 place T and R in subscript for ∆a and g ∆b

Line 147 missing ½ in equation.

Line 151 change или to or

Line 155 Eq2 and 3 are the same. Please fix it

In Table 1 values are not given in weight %, but in PHR (parts per hundred resin)

In the Materials and Methods part Authors should add more date regarding used epoxy resins and hardeners. For this Authors should use manufacturers MSDS and TDS but if shown necessary Authors may use basic techniques to determine main components of used resins. I strongly suggest to do this because then in the discussion part Authors may compare obtained results and see if there is any correlation between resin structure and obtained results. It is interesting to see if novel technique could be used for resins with different structure.    

Author Response

Dear Reviewer!

Thank you for your consideration and comments to our manuscript “Prediction of Durability of Thermal Insulating Epoxy Coatings with Regard to Climatic Ageing”. 

Point 1: In the Methods part, explain methods in more detail. For example for, tensile testing add speed at which samples were tested, as well as, samples geometry and number of samples.

 Response 1: Yes, we agree with your suggestion. Completed the Methods part:

The adhesion strength - As a sample of the test coating, a disk with a diameter of 30 mm was used, the tear force was increased, the force was increased at a rate of not more than 1 MPa/s, perpendicular to the plane of the surface to be painted, so that destruction occurred within 90 s from the beginning of the application of the tear force.

The thermal conductivity coefficient:

- thermal conductivity measurement range - 0.02-1.5 W/(m·K);

- thermal resistance measurement range - 0.01-1.5 m2·K/W;

- the limits of permissible relative error in measuring thermal conductivity and thermal resistance are ±5.0%;

- thickness of the measured sample – 10-25 mm.

It is difficult to specify the full dimensionality (as a single reference) of the samples, as the test methods are different according to GOST.

The number of samples is presented in Tables 1-7, with an average of 4 to 6 samples per Table.

Point 2: Line 144 place ab in subscript in ∆Eab∗

Response 2: ∆???

The footnote is corrected.

Point 3: Line 145 place T and R in subscript for ∆a and g ∆b

 Response 3: ∆? = ?? − ?? ; ∆? = ?? − ??

The footnote is corrected.

 Point 4: Line 147 missing ½ in equation.

Response 4: ∆??? = ?? {[(∆???)2 – (∆???)2−(∆???)2]1/2}

Added, missed while working with the formula. Thanks for the painstaking review of the formulas.

 Point 5: Line 151 change или to or

 Response 5: Yes, we agree. Corrected to “or”

Point 6: Line 155 Eq2 and 3 are the same. Please fix it

Response 6: Yes, we agree. In the text equation 3 has been deleted. All numbering corrected, reference in the text corrected.

Point 7: In Table 1 values are not given in weight %, but in PHR (parts per hundred resin)

 Response 7: Yes, we agree. The epoxy compositions were made to the most convenient values. Test compositions in mass fractions.

Point 8: In the Materials and Methods part Authors should add more date regarding used epoxy resins and hardeners. For this Authors should use manufacturers MSDS and TDS but if shown necessary Authors may use basic techniques to determine main components of used resins. I strongly suggest to do this because then in the discussion part Authors may compare obtained results and see if there is any correlation between resin structure and obtained results.

Response 8:

1) Epoxy resin - ED-20 (epoxy-dian resin) dynamic viscosity at (25±0.1) oC - 12-25Pa·s, gelatinisation time at least - 4h, mass fraction of epoxy groups at least 20.0-22.5%; Etal-247 is a low-viscosity modified epoxy resin. Viscosity 20-22 times lower than ED-20, gelatinisation time at least - 1.5 h, mass fraction of epoxy groups at least 21.4-22.8%; 

2) Hardener - Etal – 1440N is a low-viscosity two-pack epoxy compound, viable for up to 3 hours, curing period - 24 hours, at 20-25 °C, workable up to 110 °C after additional heat treatment; Etal - 1460 amine type is a homogenous low-viscous liquid of light to dark brown colour, designed for curing of epoxy resins and compounds based on them at 0 - +40°C in any humidity and under water, curing time of compound at 0-10°C is 24 hours, at 10-20°C is 16 hours; Etal-45M is an amine type hardener used to harden epoxy resins and compounds based on them at temperatures from -7 oC (recommended from +5) to +45 oC, % humidity is not important; Polyethylene polyamine (PEPA) is a cold-curing hardener for epoxy resins and compositions. It is a mixture of ethylene amines. PEPA is fully soluble in water and in alcohol.

 Point 9: It is interesting to see if novel technique could be used for resins with different structure.    

Response 9:

According to GOST R 56211-2014 or GOST 10587-84 "Uncured epoxy-diane resins" there are 6 main types and 8 subspecies of resins, under the guise of certain trademarks, the type of these resins is about 25, which are used as the main component for liquid coatings. If in the resin used the name of the indicator by molecular weight, epoxy equivalent, viscosity according to a ball viscometer, viscosity of the resin with a hardener according to a ball viscometer at 100 °C corresponds to GOST, new compositions can be considered (developed).

Best Regards,

Authors group

Round 2

Reviewer 1 Report

The authors addressed all of the comments from the reviewer. Thus, it was suggested to be accepted by the Journal.

Author Response

Dear Reviewer!

Thank you for your consideration and comments to our manuscript “Prediction of Durability of Thermal Insulating Epoxy Coatings with Regard to Climatic Ageing”. 

I am sending a revised version of the manuscript .

Thank you for your support.

Bets Regards,

Zhanar O.Zhumadilova

Reviewer 2 Report

The authors considered most of the comments or adequately responded to the remarks contained in the review; therefore, the work may be approved for publication.

Author Response

(The authors gave the same response as above.)
